# Upregulated Guanine Deaminase Is Involved in Hyperpigmentation of Seborrheic Keratosis via Uric Acid Release

**DOI:** 10.3390/ijms222212501

**Published:** 2021-11-19

**Authors:** Kyung Ah Cheong, In Sup Kil, Hyuk Wan Ko, Ai-Young Lee

**Affiliations:** 1Department of Dermatology, Dongguk University Ilsan Hospital, 814 Siksa-dong, Ilsandong-gu, Goyang-si 410-773, Gyeonggi-do, Korea; bionase@hanmail.net; 2Basic Research & Innovation Division, Amorepacific Corporation R&D Center, Yongin-si 446-729, Gyeonggi-do, Korea; iskil@amorepacific.com; 3Department of Biochemistry, College of Life Science and Biotechnology, Yonsei University, Seoul 03722, Korea; kohw@yonsei.ac.kr

**Keywords:** *GDA*, photoaging, hyperpigmentation, seborrheic keratosis, uric acid

## Abstract

Seborrheic keratosis, which is a benign tumor composed of epidermal keratinocytes, develops common in the elderly. Uric acid generated by upregulated guanine deaminase (*GDA*) has been identified to cause UV-induced keratinocyte senescence in seborrheic keratosis. Seborrheic keratosis is also frequently pigmented. Growing evidences indicate that hyperuricemia is a risk factor of acanthosis nigricans, an acquired skin hyperpigmentation. The objective of this study was to investigate role of *GDA* and its metabolic end product, uric acid, in hyperpigmentation of patients with seborrheic keratosis using their lesional and non-lesional skin specimen sets and cultured primary human epidermal keratinocytes with or without *GDA* overexpression or uric acid treatment. *GDA*-overexpressing keratinocytes or their conditioned media containing uric acid increased expression levels of MITF and tyrosinase in melanocytes. Uric acid released from keratinocytes was facilitated by ABCG2 transporter with the help of PDZK1 interaction. Released uric acid was taken by URAT1 transporter in melanocytes, stimulating melanogenesis through p38 MAPK activation. Overall, *GDA* upregulation in seborrheic keratosis plays a role in melanogenesis via its metabolic end product uric acid, suggesting that seborrheic keratosis as an example of hyperpigmentation associated with photoaging.

## 1. Introduction

Seborrheic keratosis is a common benign skin tumor characterized by keratinocyte proliferation. Its predilection sites are sun-exposed areas and its affected age group is the elderly [1,2], suggesting the role of photoaging in the development of seborrheic keratosis [3,4]. We have recently found that guanine deaminase (*GDA*) expression in keratinocytes induced by repeated ultraviolet (UV) exposure is involved in UV-induced keratinocyte senescence in seborrheic keratosis via uric acid, a guanine metabolite [5], supporting a causative role of photoaging in seborrheic keratosis. Sunlight can stimulate melanin production to protect skin against harmful effects of UV radiation. Tyrosinase and tyrosinase-related proteins (TRPs) are key enzymes for melanogenesis. Genes encoding tyrosinase and TRPs contain common transcription starting sites, particularly micropthalmia-associated transcription factor (MITF) binding sites. MITF plays a fundamental role in the transcriptional regulation of melanogenesis [6]. Seborrheic keratosis frequently presents as sharply demarcated light brown to black papules due to great amount of melanin in its keratinocytes. It is considered as one of common facial pigmentary disorders of greatest concern along with melasma [7]. Although the role of uric acid in skin hyperpigmentation is unknown, hyperuricemia is well-known to be associated with metabolic syndrome [8,9,10,11,12]. Among cutaneous hyperpigmentations due to increased melanin, acanthosis nigricans characterized by velvety brownish-black thickened hyperpigmentation [13,14] is a representative disorder associated with metabolic syndrome. Growing evidence indicates that hyperuricemia is an important risk factor for acanthosis nigricans [15,16]. We assumed that uric acid generated by *GDA* could contribute to hyperpigmentation in seborrheic keratosis. Thus, the objective of this study was to examine whether and how *GDA* and its metabolic end product, uric acid, were involved in melanogenesis and skin hyperpigmentation in patients of with seborrheic keratosis.

## 2. Results

### 2.1. GDA Upregulation in Keratinocytes Stimulates Melanogenesis by Releasing Uric Acid

*GDA* mRNA and protein have been detected in keratinocytes, but not in melanocytes [5]. Thus, the role of upregulated *GDA* in melanogenesis was examined in primary cultured adult normal human keratinocytes with or without *GDA* overexpression cocultured with primary cultured adult normal human melanocytes. Levels of tyrosinase mRNA and protein were increased in normal human melanocytes cocultured with *GDA*-overexpressing keratinocytes (Figure 1a). Levels of tyrosinase and MITF proteins were also elevated in melanocyte monocultures treated with supernatant obtained from *GDA*-overexpressing keratinocytes (Figure 1b). *GDA* concentrations were increased in conditioned media from *GDA*-overexpressing keratinocytes without showing a significant effect on cell survival and cytotoxicity (Figure 1c). However, recombinant GDA did not increase tyrosinase protein levels in melanocytes (Figure 1d). Concentrations of uric acid, a metabolic end product of *GDA*, measured for 2 days in culture supernatants after *GDA* overexpression were increased with the lapse of time (Figure 1e). Exogenous uric acid increased expression levels of tyrosinase and MITF proteins in melanocytes in a dose-dependent manner (Figure 1f). Levels of tyrosinase and MITF proteins increased by conditioned media from *GDA*-overexpressing keratinocytes along with uric acid were reduced when *GDA*-overexpressing keratinocytes were treated with allopurinol, a xanthine oxidase inhibitor (Figure 1g).

### 2.2. Uric Acid Is Secreted from Keratinocytes through ABCG2

ABCG2 is one of main urate transporters for uric acid excretion expressed on apical membranes in several tissues [17], including interfollicular keratinocyte progenitor cells [18]. In the present study, expression levels of ABCG2 in epidermal keratinocytes and role of ABCG2 in the excretion of uric acid from epidermal keratinocytes were examined. Levels of ABCG2 protein, which was detectable in keratinocytes, were increased by *GDA* overexpression (Figure 2a). Treatment with Ko143 [(3S,6S,12aS)-1,2,3,4,6,7,12,12a-octahydro-9-methoxy-6-(2-methylpropyl)-1,4-dioxopyrazino[1’,2′:1,6]pyrido[3,4-b]indole-3-propanoic acid 1,1-dimethylethyl ester], an inhibitor of ABCG2 [19], reduced uric acid concentrations induced by *GDA* overexpression in keratinocyte culture supernatants, but increased them in keratinocyte lysates (Figure 2b). *ABCG2* knockdown decreased tyrosinase protein levels in normal melanocytes cocultured with *GDA*-overexpressing keratinocytes (Figure 2c). *ABCG2* knockdown reduced uric acid concentrations in culture supernatants, but increased them in cell lysates from *GDA*-overexpressing keratinocytes (Figure 2d). Immunofluorescence study for skin specimens of patients with seborrheic keratosis with stronger *GDA* staining intensities using anti-ABCG2 and anti-uric acid antibodies showed stronger staining intensities in lesional epidermis (Figure 2e).

### 2.3. ABCG2 Action Is Facilitated by PDZK1 in Keratinocytes

PDZK1 plays a pivotal role in the regulation of ABCG2 in human intestinal cell lines [20]. Therefore, the role of *GDA* in *PDZK1* expression and the role of *PDZK1* in ABCG2 regulation were examined in epidermal keratinocytes. *PDZK1* protein levels in keratinocytes were increased by *GDA* overexpression (Figure 3a). *PDZK1* overexpression increased levels of ABCG2 (Figure 3b). Immunoprecipitation assay showed a detectable binding between *PDZK1* and ABCG2 in *PDZK1*-overexpressing keratinocytes (Figure 3c). Staining intensity with an anti-ABCG2 antibody was stronger in lesional epidermis compared to normal epidermis of seborrheic keratosis showing increased *PDZK1* staining intensity (Figure 3d).

### 2.4. Uric Acid Uptake through URAT1 in Melanocytes, Stimulating Melanogenesis via p38 MAPK

Uric acid was excreted from keratinocytes through ABCG2 (Figure 3a and Figure 4d) with the help of PDZK1 (Figure 3a–c). Uric acid should be taken by melanocytes to induce melanogenesis. URAT1 is a major transporter responsible for the reabsorption of urate in kidney and from blood [21,22]. Therefore, the presence and the role of URAT1 in uric acid uptake were examined in melanocytes. Treatment with uric acid increased levels of URAT1 in primary cultured epidermal melanocytes (Figure 4a). Levels of tyrosinase and MITF proteins were also increased by uric acid (Figure 4a). However, their levels were reduced by probenecid, an inhibitor of URAT1 [23], along with intracellular uric acid concentrations (Figure 4b). *URAT1* knockdown also reduced levels of intracellular uric acid along with tyrosinase and MITF proteins in melanocytes treated with exogenous uric acid (Figure 4c). These increase of tyrosinase and MITF proteins after the application of supernatants from *GDA* overexpressing keratinocytes were also reduced by *URAT1* knockdown (Figure 4d). Immunofluorescence study for skin specimens of seborrheic keratosis using anti-URAT1 and anti-cKIT, a receptor tyrosine kinase expressed in melanocytes [24,25], antibodies showed stronger staining intensities in lesional melanocytes (Figure 4e). Uric acid activated p38 mitogen-activated protein kinase (MAPK), but not ERK or JNK, in melanocytes in a dose-dependent manner (Figure 4f). SB203580, a p38 MAPK inhibitor, reduced levels of tyrosinase and MITF proteins in the presence of uric acid (Figure 4g).

## 3. Discussion

This study first examined whether *GDA* upregulation could be involved in skin hyperpigmentation. Increased levels of tyrosinase and MITF by coculturing with *GDA*-overexpressing keratinocytes (Figure 1a) suggested a role of *GDA* in enhancing melanogenesis. In fact, *GDA* upregulation has been reported in melasma [26,27,28], a representative pigmentary skin disorder, in addition to seborrheic keratosis [5]. We have also detected upregulation of *GDA* in melasma (data not shown). Hyperpigmentation is a common clinical finding of seborrheic keratosis and melasma. In addition, UV radiation is one of main causative factors of melasma [29,30,31,32,33]. Skin aging is also emerging as a cause of melasma [31,32,34,35], indicating that seborrheic keratosis shares certain causes with melasma. The identified role of *GDA* upregulation in UV-induced keratinocyte senescence [5] suggested that upregulated *GDA* contributed to the hyperpigmention, particularly related to photoaging.

Based on the result that *GDA* upregulation stimulated melanogenesis, its mechanism was then investigated. Melanogenesis in melanocytes was also enhanced by conditioned media obtained from *GDA*-overexpressing keratinocytes (Figure 1b), suggesting the presence of *GDA* or other factors for stimulating melanogenesis in conditioned media. In fact, *GDA* level was increased in the conditioned media in the absence of keratinocyte damage (Figure 1c). However, receptor of *GDA* has not been identified in melanocytes yet. There was no enhancement of melanogenesis from melanocytes by recombinant *GDA* either (Figure 1d), suggesting roles of melanogenic factors other than *GDA* released from *GDA*-overexpressing keratinocytes in melanogenesis. An interaction between melanogenic growth factors and *GDA* in keratinocytes has been mentioned [28]. We also found that levels of melanogenic growth factors such as basic fibroblast growth factor and SCF were increased in keratinocytes by *GDA* over expression (Appendix A). Although the exact mechanism involved in the upregulation of melanogenic growth factors by *GDA*-overexpressing keratinocytes was not clarified, this study was focused on the role of *GDA* in melanogenesis related to photoaging through its metabolic end product based on the role of uric acid in UV-induced keratinocyte senescence [5]. Increased uric acid levels in culture supernatants of *GDA*-overexpressing keratinocytes which stimulated melanogenesis (Figure 1e) could support the need for investigating the role of *GDA* via uric acid. Levels of MITF and tyrosinase enhanced by exogenous uric acid (Figure 1f). Increased uric acid levels along with MITF and tyrosinase levels in *GDA*-overexpressing keratinocytes were restored by allopurinol (Figure 1g). These findings indicate a significant role of uric acid in GDA-induced melanogenesis.

Although keratinocytes generated uric acid (Figure 1e), how uric acid was released from keratinocytes was unclear. Therefore, the role of a transporter for uric acid excretion, ABCG2 [17], was examined. ABCG2 levels were upregulated in GDA-overexpressing keratinocytes (Figure 2a). Ko143, an ABCG2 inhibitor, and ABCG2 knockdown reduced GDA-induced uric acid concentrations in supernatants but increased them in cell lysates (Figure 2b–d). These results indicated the necessity of ABCG2 upregulation for excretion of uric acid in GDA-overexpressing keratinocytes. In addition, GDA-induced tyrosinase levels were reduced by ABCG2 knockdown (Figure 2c), indicating that ABCG2 was involved in melanogenesis by regulating uric acid excretion. Stronger staining intensities with anti-ABCG2 and anti-uric acid antibodies in lesional skin of seborrheic keratosis with GDA upregulation (Figure 2e) supported an association between ABCG2 and uric acid in seborrheic keratosis. Although the regulatory role of PDZK1 in ABCG2 [20] has not been identified in the skin, PDZK1 was upregulated in keratinocytes by GDA overexpression (Figure 3a). In addition, ABCG2 levels were enhanced by PDZK1 overexpression (Figure 3b) and the binding between ABCG2 and PDZK1 (Figure 3c), indicating that the role of ABCG2 in GDA-induced melanogenesis could be facilitated by PDZK1. PDZK1 might play a role as a scaffold as shown in estrogen-related skin pigmentation [36]. Stronger staining intensities with anti-PDZK1 and anti-ABCG2 antibodies in lesional epidermis of seborrheic keratosis (Figure 3d) also supported clinical implications of their relations.

In order to participate in melanogenesis, uric acid excreted from keratinocytes should be taken by melanocytes. It has been reported that URAT1 is a transporter for uric acid resorption [21,22]. Although such data are unavailable for the skin, levels of URAT1 were enhanced by uric acid in a dose-dependent manner (Figure 4a) while levels of tyrosinase, MITF as well as intracellular uric acid were reduced by a URAT1 inhibitor (Figure 4b) or URAT1 knockdown (Figure 4c,d), indicating an important role of URAT1 in uric acid-stimulated melanogenesis by uric acid reabsorption in melanocytes. Through continuing efforts to find URAT1 inhibitors to control hyperuricemia more safely, some natural products such as nobiletin (polymethoxyflavonoid), hesperetin, naringenin (flavanones), and dioscin (steroidal saponin) have been identified as materials suitable for hyperuricemia control [37,38]. Although a few studies have examined effects of these natural products on melanogenesis inhibition, result from some of these URAT1 inhibitors [39,40,41,42,43] supported a role of URAT1 in melanogenesis. Stronger staining intensity with an anti-URAT1 antibody in lesional melanocytes, which were identified by positive staining with anti-cKIT antibody, from patients with seborrheic keratosis (Figure 4e) reinforced a role of URAT1 in hyperpigmentation of seborrheic keratosis.

Finally, how the uric acid taken by melanocytes stimulated melanogenesis was examined. MAPK is one of melanogenesis-related signaling pathways [44,45,46]. Activation of p38 MAPK is presented in the signaling pathway for uric acid action [47,48,49]. Examining the role of MAPK in uric acid induced melanogenesis showed an activation of p38 MAPK by uric acid (Figure 4f) and a reduction of uric acid-induced melanogenesis by a p38 MAPK inhibitor (Figure 4g). These results indicate that uric acid could stimulate melanogenesis through p38 MAPK signaling pathway.

In summary, *GDA* upregulation in keratinocytes can stimulate melanogenesis via uric acid generation. Uric acid is then excreted by the upregulation and interaction be-tween ABCG2 and PDZK1. Excreted uric acid is then reabsorbed by melanocytes via URAT1 upregulation, leading to melanogenesis through p38 MAPK activation (Figure 5). Considering that *GDA* is involved in UV-induced keratinocyte senescence, these events support the association of hyperpigmentation with photoaging in seborrheic keratosis.

Excretion of uric acid is induced by the upregulation and interaction between ABCG2 and PDZK1 in *GDA*-overexpressing keratinocytes. Excreted uric acid is reabsorbed by melanocytes via URAT1 upregulation through p38 MAPK activation, leading to melanogenesis.

## 4. Materials and Methods

### 4.1. Patients

Four patients diagnosed with seborrheic keratosis with a mean age of 64.5 were included in the study. This study was approved by the Institutional Review Board of Dongguk University Ilsan Hospital (the approval No. 2012-01-032). It was conducted according to the Declaration of Helsinki Principles. After obtaining informed written consent from each patient, pairs of hyperpigmented and adjacent normally pigmented skin specimens taken for biopsy were used for direct comparisons through real-time PCR and immunohistochemistry.

### 4.2. Normal Human Epidermal Cell Culture

Adult skin specimens obtained from Caesarean sections and circumcisions were used to establish cell culture for keratinocytes and melanocytes [5,50,51]. Keratinocyte cultures were obtained by suspending individual epidermal cells derived according to published methods, in EpiLife Medium supplemented with bovine pituitary extract (BPE), bovine insulin (BI), hydrocortisone, human epidermal growth factor, and bovine transferrin (BT) (Thermo Fisher Scientific, Waltham, MA, USA). Melanocyte cultures were obtained by suspending individual epidermal cells in Medium 254 supplemented with BPE, fetal bovine serum (FBS), BI, hydrocortisone, bFGF, BT, heparin, and phorbol 12-myristate 13-acetate (Thermo Fisher Scientific, Waltham, MA, USA). Keratinocytes between passages 3 and 5 and melanocytes at passages between 8 and 25 were used for real-time PCR, western blot, immunoprecipitation, 3-(4,5-dimethylthiazol-2yl)-2,5-diphenyltetrazolium bromide (MTT; Dongin Biotech. Seoul, Korea) assay, and lactate dehydrogenase (LDH; Dongin Biotech) assay with or without gene transfection.

### 4.3. Overexpression of *GDA* and Knockdown of ABCG2, or URAT1

Keratinocytes or melanocytes were transfected with 25 nM CRISPR-CAS9 sgRNA for human ABCG2, URAT1, or a negative control sgRNA (Integrated DNA Technologies, San Diego, CA, USA) using a CRISPRMAX transfection reagent (Thermo Fisher Scientific, Waltham, MA, USA). pCMV vector containing *GDA* gene [5] was transfected into cells using Lipofectamine 2000 (Thermo Fisher Scientific). Cells were used for experiments at appropriate time post-transfection.

### 4.4. Cell Viability and Cytotoxicity Test

To measure cell viability, cells were incubated with MTT for 4 h. Precipitated formazan was dissolved in dimethyl sulfoxide (DMSO). The optical density was measured at 570 nm with background subtraction at 630 nm using a spectrophotometer. LDH assay performed following the manufacturer’s instruction for measuring cell cytotoxicity.

### 4.5. Recombinant GDA Treatment

Melanocytes (2 × 105 cells/35 mm diameter well with an area of ~ 9.6 cm^2^) were treated with various concentrations of recombinant *GDA* (50, 100, and 200 ng/mL) (BioVision, Milpitas, CA, USA) up to 3 days.

### 4.6. Uric Acid Assay

Concentrations of uric acid were determined fluorometically (Ex/Em = 535/587 nm) in cell lysates or culture supernatants using a uric acid assay kit (Abcam, Cambridge, UK), following the manufacturer’s instructions.

### 4.7. Treatment with Allopurinol or Ko143

Keratinocytes (2 × 105 cells/9.6 cm^2^) were transfected with *GDA*. Four hours later, cells were treated with different concentrations of allopurinol (1 and 10 μM) or Ko143 (1 and 2 μM) (Sigma-Aldrich, St. Louis, MO, USA) for 2 days or 24 h, respectively. Cells or culture supernatants were harvested for melanocyte culture or uric acid assay.

### 4.8. Uric Acid Treatment with/without Probenecid

Melanocytes (2 × 105 cells/9.6 cm^2^) were treated with appropriate concentrations of uric acid (Sigma-Aldrich) with or without probenecid (20 and 40 μM) (Sigma-Aldrich) for 24 h. Cells or culture supernatants were harvested for uric acid assay.

### 4.9. Real-Time PCR

cDNA was synthesized from the total RNA using the cDNA Synthesis Kit for RT-PCR (Promega, Fichburg, WI, USA). The amount of target mRNA was quantified by real-time PCR using a Light Cycler real-time PCR machine (Roche, Penzberg, Germany). The relative amount of mRNAs was calculated as the ratio of each target relative to glyceraldehyde 3-phosphate dehydrogenase (GAPDH). Primer sequences used were as follows: *GDA*, 5′-catagtgacaccacgtttttcc-3′(Forward) and 5′-cgattttcacttatatggctctga-3′ (Reverse); Tyrosinase, 5′-agcatcattcttctcctcttgg-3′ (Forward) and 5′-gcataaagactgatggctgttg-3′ (Reverse); GAPDH, 5′-tccactggcgtcttcacc-3′ (Forward) and 5′-ggcagagatgatgacccttt-3′ (Reverse).

### 4.10. Western Blot Analysis

Equal amounts of extracted proteins were resolved and transferred to nitrocellulose membranes. Each membrane was incubated with one of the following antibodies against ABCG2, *GDA*, PDZK1, tyrosinase, and URAT1 (mouse monoclonal; Santa Cruz Biotechnology, Dallas, TX, USA), MITF, phospho-ERK, ERK, phospho-p38, p38, and JNK (rabbit polyclonal; Cell Signaling Technology, Beverly, MA, USA), phospho-JNK (mouse monoclonal; cell signaling technology) and β-Actin (mouse monoclonal; Sigma-Aldrich). After incubating with appropriate anti-mouse or anti-rabbit horseradish peroxidase-conjugated antibodies (Thermo Fisher Scientific), an enhanced chemiluminescence solution (Thermo Fisher Scientific) was applied and signals were captured with an image reader (LAS-3000; Fuji Photo Film, Tokyo, Japan). The protein bands were then analyzed by densitometry.

### 4.11. Immunoprecipitation

Supernatants of cell lysates were incubated with an anti-PDZK1 antibody and resin of Pierce™ Direct IP kit (Thermo Fisher Scientific) at 4 °C. Eluted resin-bound proteins were analyzed by immunoblotting with anti-ABCG2 or anti-ABCG2 antibody.

### 4.12. Immunohistochemistry

After deparaffinization and rehydration, sections were preincubated with 3% bovine serum albumin. These sections were reacted sequentially with the an anti-*GDA*, anti-PDZK1, or anti-URAT1 antibody and 1:200 Alexa Fluor-labeled goat anti-mouse IgG (488; Molecular Probes, Eugene, OR, USA), with anti-ABCG2 and Alexa Fluor-labeled goat anti-mouse IgG (594; Molecular Probes), or with anti-cKit (Cell signaling technology), anti-uric acid (Santa Cruz Biotechnology) antibody, and Alexa Fluor-labeled goat anti-rabbit IgG (594; Molecular Probes). Nuclei were counterstained with Hoechst 33258 (Sigma-Aldrich). Fluorescence images were evaluated using an image analysis system (Dp Manager 2.1; Olympus Optical Co., Tokyo, Japan) and a Wright Cell Imaging Facility (WCIF) ImageJ software 1.53e (http://www.uhnresearch.ca/facilities/wcif/imagej accessed on 9 August 2021).

### 4.13. Statistical Analysis

All statistical analyses were performed using GraphPad Prism 5.03 (GraphPad Software, La Jolla, CA, USA). A *p*-value of <0.05 was considered significant. Statistical comparisons between two groups were performed using two-tailed Student’s unpaired *t*-test (parametrical data). For comparisons among multiple groups, one-way Analysis of Variance (ANOVA) was used. For human sample data, differences between non-lesions and lesions were assessed with the Mann–Whitney U test. Values of mean ± SD were calculated for in vitro experimental data.

## 5. Conclusions

*GDA* upregulation stimulated melanogenesis. Regarding the mechanism of *GDA*-induced hyperpigmentation, *GDA* upregulation enhanced melanogenesis via uric acid generation. More uric acid was excreted by *GDA*-overexpressing keratinocytes through ABCG2 with the help of PDZK1. Excreted uric acid, which was reabsorbed by melanocytes via URAT1 upregulation, enhanced melanogenesis through p38 MAPK activation. Although other factors could be involved in *GDA*-induced hyperpigmentation, generation of uric acid, a metabolic end product, could contribute to *GDA*-induced-melanogenesis.

## 6. Patents

It was described in the Materials and Methods of the original.

## Figures and Tables

**Figure 1 ijms-22-12501-f001:**
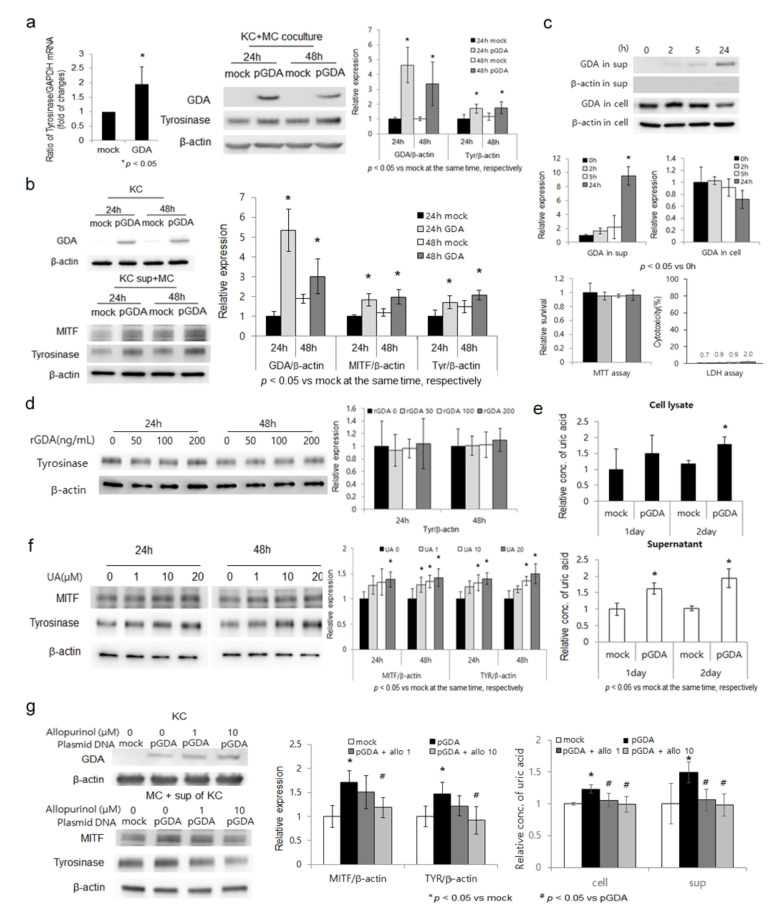
*GDA* upregulation in keratinocytes stimulates melanogenesis by releasing uric acid. (**a**) Real-time PCR and Western bot analyses for relative ratios of tyrosinase mRNA and protein levels, respectively, in primary cultured normal human skin melanocytes (MC) cocultured with *GDA*-overexpressing keratinocytes (KC). (**b**) Western blot analysis to determine relative ratios of tyrosinase and MITF protein levels in normal melanocytes treated with supernatants from *GDA*-overexpressing keratinocytes. (**c**) Western blot analysis for relative ratios of *GDA* protein levels in culture supernatants and *GDA*-overexpressing keratinocytes, whose viability and cytotoxicity were measured by MTT and LDH assays, respectively. (**d**) Western blot analysis for relative ratios of tyrosinase in normal melanocytes treated with recombinant *GDA* (rGDA). (**e**) Uric acid assay for measuring concentrations of uric acid in cell lysates and culture supernatants from keratinocytes after *GDA* overexpression for 2 days. (**f**) Western blot analysis for relative ratios of tyrosinase and MITF proteins in normal melanocytes treated with various concentrations of uric acid (UA). (**g**) Western blot analysis for relative ratios of tyrosinase and MITF proteins in normal melanocytes treated with supernatants from *GDA*-overexpressing keratinocytes cultured in the presence of various concentrations of allopurinol along with uric acid concentrations. β-Actin and GAPDH were used as internal controls for Western blot analysis and real-time PCR, respectively. Data represent means ± SD of four independent experiments.

**Figure 2 ijms-22-12501-f002:**
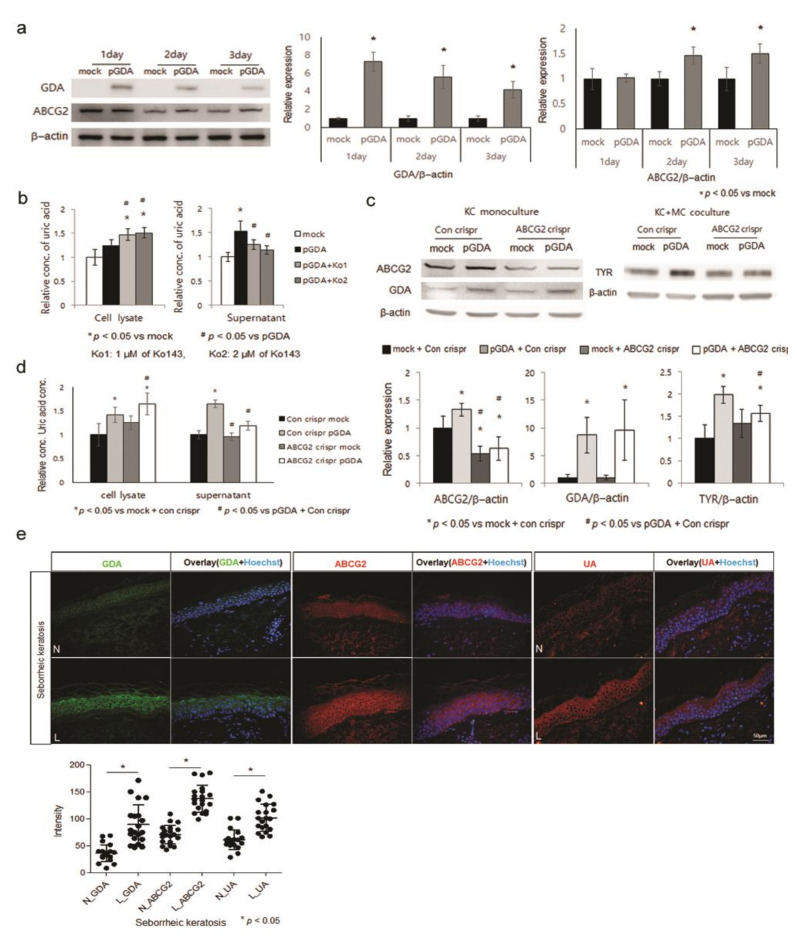
Uric acid is secreted from keratinocytes through ABCG2. (**a**) Western blot analysis for relative ratios of ABCG2 protein levels in keratinocytes after *GDA* overexpression. (**b**) Uric acid assay determining concentrations of uric acid in cell lysates and culture supernatants from *GDA*-overexpressing keratinocytes in the presence of Ko143 at different concentrations up to 24 h. (**c**) Western blot analysis for relative ratios of tyrosinase protein levels in melanocytes (MC) applied with supernatants of *GDA*-overexpressing keratinocytes (KC) with or without *ABCG2* knockdown. (**d**) Uric acid assay for concentrations of uric acid in cell lysates and culture supernatants from *GDA*-overexpressing keratinocytes with or without *ABCG2* knockdown. (**e**) Representative immunofluorescent staining using anti-GDA and anti-ABCG2 antibodies in lesional (L) skin samples compared to that in non-lesional (N) skin samples from four patients with seborrheic keratosis. Nuclei were counter-stained with Hoechst 33258 (Bar = 0.05 mm). Staining intensities were measured for five randomly selected areas in each image using a Wright Cell Imaging Facility ImageJ software. β-Actin was used as an internal control for Western blot analysis. Data represent means ± SD of four independent experiments.

**Figure 3 ijms-22-12501-f003:**
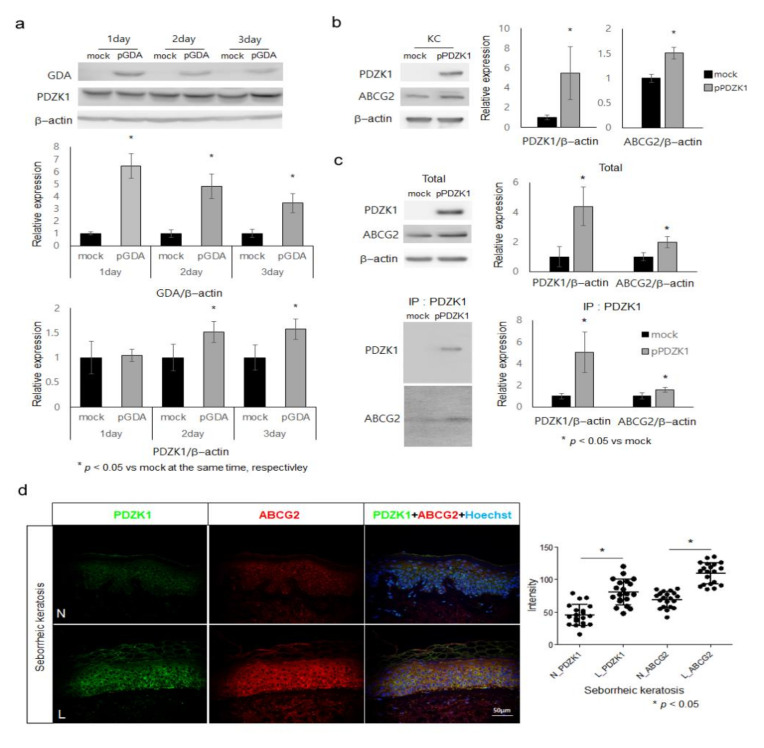
ABCG2 action is facilitated by PDZK1 in keratinocytes. (**a**) Western blot analysis for relative ratios of PDZK1 protein levels in keratinocytes after *GDA* overexpression. (b) Western blot analysis for relative ratios of ABCG2 protein levels in keratinocytes (KC) with *PDZK1* overexpression. (**c**) Immunoprecipitation using anti-PDZK1 and anti-ABCG2 antibodies in cultured keratinocytes with or without *PDZK1* overexpression. (**d**) Representative immunofluorescent staining using anti-PDZK1 and anti-ABCG2 antibodies in lesional (L) skin samples compared to that in non-lesional (N) skin samples from four patients with seborrheic keratosis. Nuclei were counter-stained with Hoechst 33258 (Bar = 0.05 mm). Staining intensities were measured for five randomly selected areas in each image using a Wright Cell Imaging Facility ImageJ software. β-Actin was used as an internal control for Western blot analysis. Data represent means ± SD of four independent experiments.

**Figure 4 ijms-22-12501-f004:**
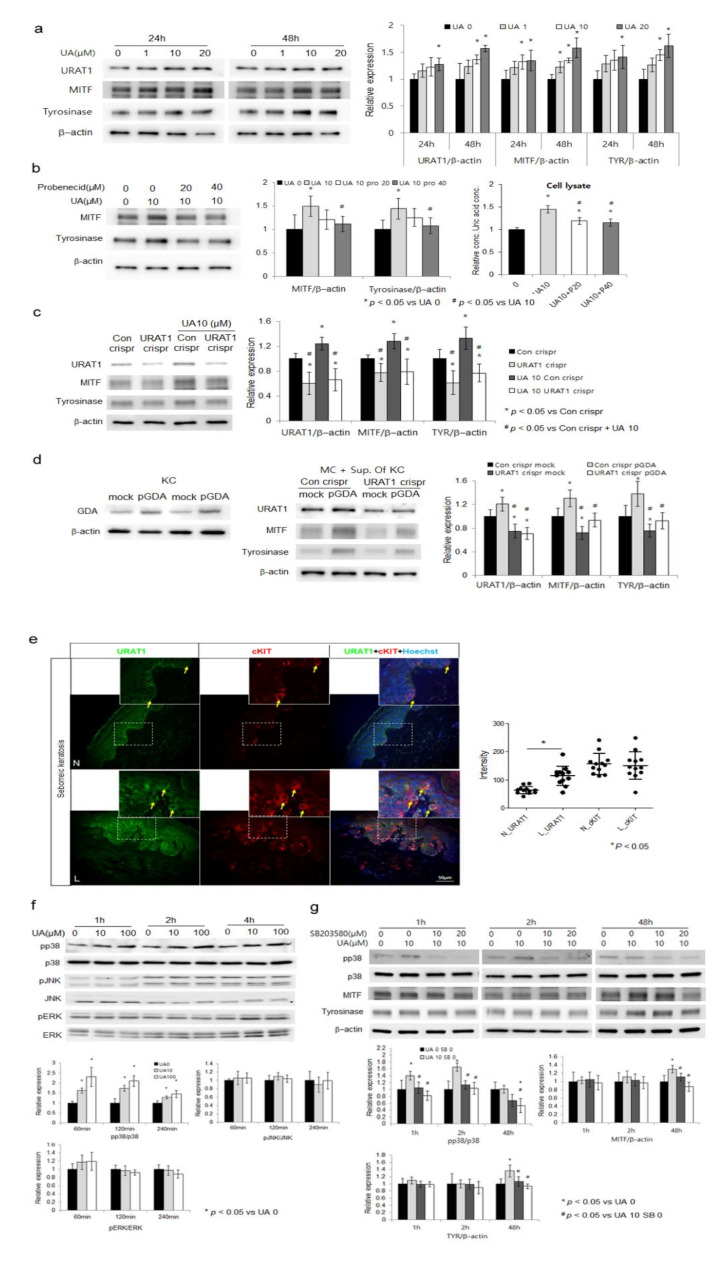
Uric acid uptake through *URAT1* in melanocytes, stimulating melanogenesis via p38 MAPK. (**a**) Western blot analysis for relative ratios of *URAT1*, tyrosinase, and MITF proteins in normal melanocytes treated with different concentrations of uric acid (UA). (**b**–**c**) Relative ratios of tyrosinase and MITF proteins and intracellular uric acid concentrations in normal melanocytes treated with different concentrations of probenecid (**b**) and in control or *URAT1*-knockdown melanocytes (**c**) in the absence or presence of exogenous uric acid based on Western blot analysis. (**d**) Relative ratios of tyrosinase and MITF proteins in control or *URAT1*-knockdown melanocytes (MC) treated with supernatants from *GDA*-overexpressing keratinocytes (KC) by Western blot analysis. (**e**) Representative immunofluorescent staining using anti-cKIT and anti-*URAT1* antibodies in lesional (L) skin samples compared to that in non-lesional (N) skin samples from three patients with seborrheic keratosis. Nuclei were counter-stained with Hoechst 33258 (Bar = 0.05 mm). Staining intensities were measured for four randomly selected areas in each image using a Wright Cell Imaging Facility ImageJ software. (**f**,**g**) Western blot analysis for relative ratios of p38 MAPK, JNK, and ERK proteins in normal melanocytes treated with different concentrations of uric acid in the absence (**f**) or presence of SB203580 (**g**). β-Actin was used as an internal control for Western blot analysis. Data represent means ± SD of four independent experiments.

**Figure 5 ijms-22-12501-f005:**
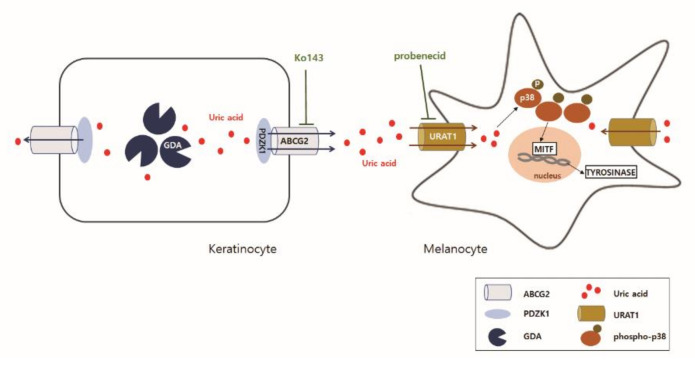
Schematic view of *GDA*-induced melanogenesis via uric acid.

## Data Availability

Not applicable.

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
