# Peer review of "Upregulated Guanine Deaminase Is Involved in Hyperpigmentation of Seborrheic Keratosis via Uric Acid Release"

_ijms, 2021, doi:10.3390/ijms222212501_

Round 1

Reviewer 1 Report

Dear Authors

The manuscript is interesting however in my opinion it requires some modification

The discussion should be modified and include concise summary of the investigated problem and critical discussion of major and minor findings . Some parts of the discussion concentrates mostly on data that are not shown e.g.: upregulation of GDA in melasma or basic fibroblast growth factor and SCF .

The conclusions must be added .

Reviewer 2 Report

The manuscript “Upregulated guanine deaminase is involved in hyperpigmentation of seborrheic keratosis via uric acid release” by Cheong et al.  aims to investigate whether GDA and uric acid are involved in hyperpigmentation of patients  with seborrheic keratosis and to obtain information regarding the mechanism. They have used cocultures of human melanocytes and keratinocytes  (with or without GDA overexpression or uric acid treatment). The Authors provide evidence that uric acid release from keratinocytes is facilitated by ABCG2 transporter, which interacts with PDZK1. Released uric acid can be transported into melanocytes through URAT1, where it  stimulates melanogenesis through p38 MAPK activation. In addition they have examined the expression of GDA, ABCG2, PDZK1, URAT1 and cKit in skin samples (lesioned and not-lesioned) of patients with seborrheic keratosis.  The results are very interesting, and the manuscript, in general, is clearly written and the co-cultures experiments are well discussed but I think that a minor revision could improve the manuscript.

  • In order to better understand the results, I think that a sentence indicating the role of tyrosinase and MITF1 in melanogenesis should be included in the introduction. Another sentence (in Results or Discussion) should be added regarding the expression of ckit in keratinocytes and melanocytes and its role on melanogenesis or seborrheic keratosis.
  • The immunohistochemical experiments should be better discussed.
  • Some details should be added.

Normal human epidermal cell culture: Please provide more information on cell culture experimental conditions (source of the cells, growth medium, etc) and include references; information on the number of cells used in the transfection experiments and  viability assays should also be provided. In some experiments there is some information [L 307, L316, L32, Melanocytes (2×105 cells/well)] but the diameter of the plate should be provided, or density should be expressed as number of cells/cm2 .

In rappresentative blots there is only one control of loading. From the description of the methods, it is not clear whether the different antibodies were applied to the same membrane or if they were applied to different membranes and only one control of loading is shown for simplicity. Were blots stripped and reused for different antibodies? If so, stripping conditions should be added to Methods. Please clarify.

  • These sentences are not clear:

 L 244 “while levels of tyrosinase and MITF with intracellular uric acid were reduced by a URAT1 inhibitor (Figure 4b) or URAT1 knockdown (Figures 4c and 4d)”.

Do you mean “while  tyrosinase, MITF as well as intracellular uric acid levels  were reduced by…”

L 255 “reinforced a role of URAT1 in hyperpigmentation of both diseases”. Which diseases? Figure refers to seborrheic  keratosis. Please specify.

  • Some typos should be corrected. For example:

L122, 146, 184:  non-leisonal

L 205 enhancment

L272 activation.; leading: activation, leading

Round 2

Reviewer 1 Report

The updated  manuscript is much improved.  I have no further comments.